# Role of Cytokines and Chemokines in Angiogenesis in a Tumor Context

**DOI:** 10.3390/cancers14102446

**Published:** 2022-05-16

**Authors:** Mannon GEINDREAU, Mélanie BRUCHARD, Frédérique VEGRAN

**Affiliations:** 1Université de Bourgogne Franche-Comté, 21000 Dijon, France; mannon.geindreau@u-bourgogne.fr (M.G.); melanie.bruchard@u-bourgogne.fr (M.B.); 2CRI INSERM UMR1231 ‘Lipids, Nutrition and Cancer’ Team CAdiR, 21000 Dijon, France; 3Centre Georges-François Leclerc, UNICANCER, 21000 Dijon, France; 4LipSTIC Labex, 21000 Dijon, France

**Keywords:** cancer, angiogenesis, therapy, cytokines, chemokines

## Abstract

**Simple Summary:**

Tumor growth in solid cancers requires adequate nutrient and oxygen supply, provided by blood vessels created by angiogenesis. Numerous studies have demonstrated that this mechanism plays a crucial role in cancer development and appears to be a well-defined hallmark of cancer. This process is carefully regulated, notably by cytokines with pro-angiogenic or anti-angiogenic features. In this review, we will discuss the role of cytokines in the modulation of angiogenesis. In addition, we will summarize the therapeutic approaches based on cytokine modulation and their clinical approval.

**Abstract:**

During carcinogenesis, tumors set various mechanisms to help support their development. Angiogenesis is a crucial process for cancer development as it drives the creation of blood vessels within the tumor. These newly formed blood vessels insure the supply of oxygen and nutrients to the tumor, helping its growth. The main factors that regulate angiogenesis are the five members of the vascular endothelial growth factor (VEGF) family. Angiogenesis is a hallmark of cancer and has been the target of new therapies this past few years. However, angiogenesis is a complex phenomenon with many redundancy pathways that ensure its maintenance. In this review, we will first describe the consecutive steps forming angiogenesis, as well as its classical regulators. We will then discuss how the cytokines and chemokines present in the tumor microenvironment can induce or block angiogenesis. Finally, we will focus on the therapeutic arsenal targeting angiogenesis in cancer and the challenges they have to overcome.

## 1. Introduction

There are two fundamental processes to form blood vessels: vasculogenesis and angiogenesis. Vasculogenesis corresponds to the de novo blood vessel formation, whereas angiogenesis is the formation of new blood vessels from pre-existing vessels. Angiogenesis is required in physiological processes such as embryogenic development and the menstrual cycle. This mechanism is also widely involved in cancer development. The involvement of this process in cancer began to be highlighted in 1800. Indeed, German researchers observed that some tumors were richly vascularized, suggesting that new blood vessel formation happened in some cancers. Later, in 1948, Algire demonstrated in mice that melanoma growth is preceded by blood vessel development [1]. Progressively, research better defined angiogenesis, in 2004 the first anti-angiogenic treatment was approved as a cancer treatment and Hanahan and Weinberg identified it as a hallmark of cancer [2].

The formation of new blood vessels from preexisting vessels is achieved in sequential steps (Figure 1). In a hypoxic environment, angiogenic factors bind to their receptor, present at the surface of endothelial cells, promoting their dilatation and activation. Simultaneously, hypoxia upregulates the expression of some proteases that induce basement membrane degradation and pericytes detachment. Then, tip cells, which are highly motile endothelial cells, migrate following the angiogenic stimuli. Thereafter, endothelial cells proliferate, inducing the formation of new blood vessels. At a later stage, the basement membrane is reformed and pericytes are recruited. Finally, the different blood vessels merge, concluding the formation of the tumor vasculature [3].

This mechanism is tightly regulated by a balance between pro and anti-angiogenic factors and is mainly induced by hypoxia, which promotes an imbalance in favor of pro-angiogenic factors. Indeed, hypoxia induces HIF-1α accumulation which in turn induces the release of pro-angiogenic factors such as vascular endothelial growth factor (VEGF) [4]. In short, when the tumor reaches approximately 2 mm, the tumor environment becomes hypoxic, inducing an angiogenic switch and the triggering of angiogenesis [5].

Cytokines and chemokines are soluble proteins able to act remotely on cells and tissues. They act on target cells by binding to specific high-affinity receptors. In this review, we will focus on the role of cytokines and chemokines in the modulation of angiogenesis in a tumor context. Finally, we will evaluate how therapies can modulate tumor angiogenesis.

**Figure 1 cancers-14-02446-f001:**
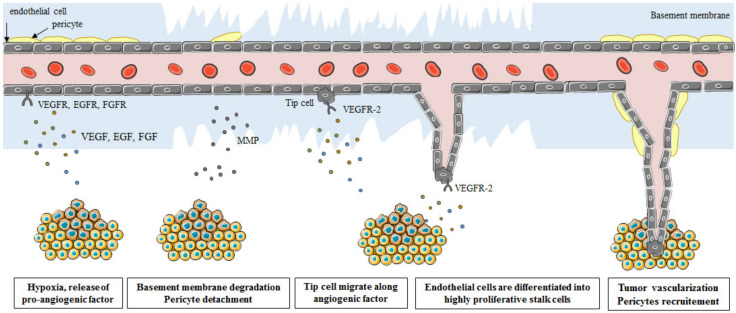
Neoangiogenesis in tumor. A tumor needs nutrients and oxygen (O_2_) to support neoplastic expansion. The provision of these needs requires the establishment of a new vascular network through the process of angiogenesis. Angiogenesis consists of the assembly of endothelial cells in the form of tubes from existing vessels. During hypoxia and tumor growth, the nuclear translocation of HIF1α induces the expression of pro-angiogenic factors such as VEGF, EGF, or FGF... Angiogenic factors are able to activate and stimulate endothelial cells through membrane receptors. Indeed, these signals participate in the proliferation, invasion, migration, survival, and increase in the permeability of the vessels. Inspired from the Cancer Research Product Guide Edition 3, 2015.

## 2. Classical Regulators of Angiogenesis

### 2.1. Vascular Endothelial Growth Factor (VEGF) Family

The most important inducer of angiogenesis is the VEGF family. The VEGF family consists of five members: VEGF-A, VEGF-B, VEGF-C, VEGF-D, and placental growth factor (PlGF). Their biological functions are mediated by three receptors: VEGFR-1, VEGFR-2, VEGFR-3, and 2 co-receptors: neuropilin and heparan sulfate proteoglycans. While VEGF-B, PlGF, and VEGF-A bind to VEGFR-1, VEGF-A, VEGF-C, and VEGF-D bind to VEGFR-2, VEGF-C, and VEGF-D bind to VEGFR-3.

The VEGF-A/VEGFR-2 signaling pathway plays a crucial role in physiological and pathological angiogenesis. Mice with VEGF-A or VEGFR-2 deficiencies are not viable and present an early embryogenic lethality due to abnormal vascular development. The VEGF family has a mitogenic and anti-apoptotic effect on endothelial cells and also induces their migration and proliferation. Furthermore, these growth factors promote blood vessel permeabilization for remodeling blood and lymphatic vessels. The expression of VEGF is enhanced by HIF-1α, the activation of oncogenes such as Ras, growth factors, and cytokines such as IL-1, Tumor Necrosis Factor-α (TNFa), Epidermal Growth Factor (EGF), and Platelet-Derived Growth Factor (PDGF). Various cells produce VEGF-A such as smooth muscle cells, keratinocytes, endothelial cells, platelets, neutrophils, and macrophages. It is believed that approximately 60% of all tumors secrete this molecule. There are many reviews on the VEGF family, so we decided not to go into details (Figure 2).

The biological function of PlGF is mediated through VEGFR-1 and two co-receptors, neuropilin-1 and neuropilin-2 [6,7]. This growth factor directly induces angiogenesis by increasing tumor vascularity and blood vessel growth and promotes survival, proliferation, and migration of endothelial cells [8,9,10]. In endothelial and vascular cells, the overexpression of HIF-1α induces the expression of PlGF [11]. Contrary to VEGF, PlGF is not required for embryogenic vessel formation but contributes to pathological angiogenesis. Indeed, mice lacking PlGF develop normal blood vessels but tumor growth and angiogenesis are reduced [12]. On contrary, Yang et al. have shown that T241 and LLC, two tumor mice models genetically modified to overexpress PlGF, present a slower tumor growth. Furthermore, these tumors present a low density of microvessels, and blood vessels are normalized. Interestingly, they also demonstrated that T241-VEGF-null cells, overexpressing PlGF, grew faster in mice, suggesting that PlGF promotes tumor growth in cells lacking VEGF expression [13]. Recently, it has been shown that PlGF is secreted by Th17 cells in vitro and in vivo. In turn, PlGF regulates Th17 differentiation through a STAT3-dependent pathway and is able to replace IL-6 functions in th17 differentiation [14].

### 2.2. The Ang-Tie System

Angiopoietins (Ang) stimulate angiogenesis and control vascular remodeling and maturation. There are four members: Ang-1 and Ang-2 are well characterized but less is known about the two others: Ang-4 and its mouse ortholog, Ang-3 [15,16,17]. Their biological functions are mediated by two receptors: Tie-1 and Tie-2 which are nearly exclusively expressed in the endothelium but also in some hematopoietic cells [18,19]. Tie-1 is an orphan receptor, meaning that it is activated by angiopoietin through its interaction with Tie-2 [20]. This system plays a crucial role in angiogenesis. Indeed, mice with Ang-1, Tie-1, or Tie-2 deficiencies present an abnormal vascular system resulting in embryonic lethality [21,22]. However, in mice deficient in Ang-2, developmental angiogenesis is mostly unaffected but results in newborns with lymphatic dysfunction and sometimes postnatal death due to chylous ascites [23]. Interestingly, Ang-2 overexpression causes embryonic lethality [17]. Ang-2 can act as an agonist or antagonist of Tie-2, depending on the context. When Ang-2 acts as an antagonist, it induces vascular destabilization and leakiness leading to vascular regression [24]. Under normal conditions, blood vessels are stable and quiescent. Ang-2 is stored in Weibel-Palade bodies [25] and its expression is low. Ang-1 suppresses Ang-2 transcription and its expression dominates. Ang-1 is a constitutive agonist of Tie-2. This molecule is expressed by mural cells, fibroblasts, tumor cells, and non-vascular cells [26]. The Ang-1/Tie-2 signaling pathway increases vascular stability and inhibits vascular permeability by acting on the EC-EC junction and on the actin cytoskeleton [27]. Under inflammatory conditions, Ang-2 is upregulated and competes with Ang-1 for binding to Tie-2. Ang-2 is rapidly released from endothelial cells and its effects are amplified by cytokines such as TNF-α and VEGF [28,29]. During inflammation, Ang-2 switches to an antagonist function and this mechanism depends on the Tie-1 receptor cleavage [30,31]. Ang-2 is highly expressed in many types of tumors such as melanoma, RCC, glioblastoma, breast, and colorectal cancer [32,33,34,35] and Ang-2 deficient mice show a reduced tumor growth in metastatic colony formation in the lung [36] (Figure 2).

### 2.3. Hepatocyte Growth Factor (HGF)

This growth factor is commonly produced by stromal cells such as fibroblasts and also by colorectal and breast cancer cells due to HGF promoter region mutations [37,38,39]. HGF is secreted in an inactive proform (pro-HGF) and its activation is mainly due to proteases that are over-expressed in tumor cells [40,41]. The overexpression of HGF in colorectal cancer stages II and III is associated with poor outcome in patients [42]. This molecule contributes to angiogenesis by promoting endothelial cell growth, survival, and migration and also stimulates epithelial-mesenchymal transition (EMT) by activating its receptor, the mesenchymal-epithelial transition factor (c-MET) [43,44]. The two molecules, c-MET and HGF have an increased expression in various cancers such as non-small cell lung carcinoma (NSCLC), gastric, ovarian, pancreatic, thyroid, breast, head and neck, colon, and kidney carcinomas [45]. Furthermore, an in vitro study has demonstrated the ability of HGF to stimulate esophageal squamous cell carcinoma to express VEGF and IL-8 and to enhance the migration and invasion of cancer cells [46]. The VEGF expression induced by HGF increases angiogenesis and it has been shown that HGF can induce VEGF transcription through SP1 phosphorylation [47] (Figure 2).

### 2.4. Fibroblast Growth Factor (FGF)

The FGF family consists of 22 members: FGF-1 to FGF-23, divided into seven subfamilies: FGF1/2/5, FGF3/4/6, FGF7/10/22, FGF8/17/18, FGF9/16/20, and FGF19/21/23 and FGF 11/12/13/14 [48]. Their biological processes are mediated by four receptors: FGFR1 to 4. There is also a receptor lacking an intracellular kinase domain, FGFR5, that then acts as a coreceptor with FGFR1. These receptors are expressed by endothelial cells. The signaling pathway of FGF/FGFR regulates different biological functions such as endothelial cell proliferation, survival, differentiation, tube formation, protease production, and angiogenesis [48,49]. However, the contribution of the FGF family to angiogenesis is controversial. It has been shown that FGF-4 and FGF-8 and particularly FGF-1 and FGF-2 have pro-angiogenic properties in different models. In vitro studies have shown that FGF-2, through paracrine and autocrine mechanisms, induces the expression of VEGF by vascular endothelial cells [50]. Furthermore, the lack of FGF signaling in endothelial cells downregulates the expression of VEGFR-2 mediated through the activation of Erk1/2 [51]. There are different cancer types presenting FGFR alteration such as head and neck cancer, non-small cell lung cancer, urothelial cancer, gastric cancer, and breast cancer [52] (Figure 2).

### 2.5. Platelet-Derived Growth Factor (PDGF)

The PDGF/PDGFR signaling pathway plays an important role in angiogenesis and particularly by inducing pericyte recruitment to vessels that allow vessel stability and endothelial cell survival. There are different isoforms of PDGF: PDGF-AA, PDGF-BB, PDGF-CC, PDGF-DD, and PDGF-AB. These molecules are produced by endothelial and epithelial cells and bind to three receptors: PDGFR-*αα*, PDGFR-*αβ*, and PDGFR-*ββ*. PDGFR-*αα* is activated by all PDGF ligands apart from PDGF-DD. PDGFR-*αβ* is activated by all PDGF ligands except PDGF-AA and PDGFR-*ββ* is activated by PDGF-BB and PDGF-DD. PDGFR-*β* is expressed on vascular smooth muscle cells and pericytes and PDGFR-*αβ* is expressed on endothelial cells. Alterations of these molecules are associated with poor survival, metastatic disease, and tumor angiogenesis. PDGF/PDGFR also induces the stimulation of proangiogenic factors such as VEGF and FGF, endothelial cell proliferation, and recruitment of endothelial precursor cells to vessels. In vivo and in vitro studies have shown that PDGF-D down-regulation in SW480 inhibits tumor growth, migration, and angiogenesis whereas PDGF-D up-regulation in HCT116 is associated with tumor aggressiveness [53] (Figure 2).

**Figure 2 cancers-14-02446-f002:**
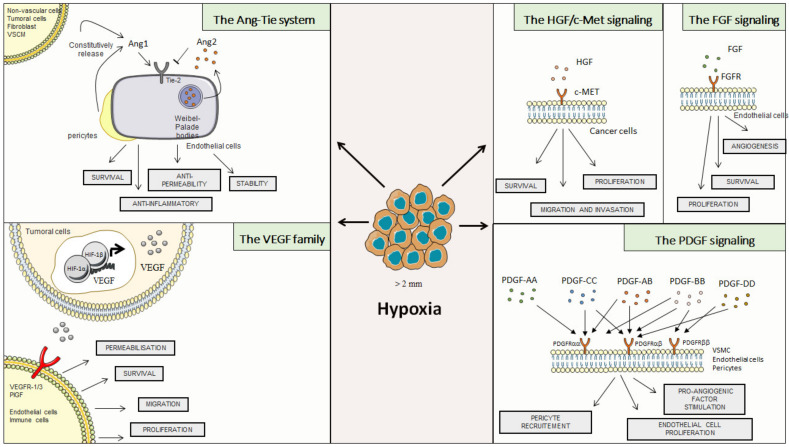
Role of classical regulators of angiogenesis.

## 3. Interleukines: A Link between the Immune System and Angiogenesis

Angiogenesis is able to modulate the immune system. This mechanism reduces immune cell infiltration by affecting the expression of proteins on endothelial cells. Angiogenesis also induces an immunosuppressive tumor microenvironment. Indeed, it induces the recruitment of immunosuppressive cells such as Treg and MDSC to the tumor, while it reduces DC maturation and CD3^+^ proliferation and cytotoxicity. Conversely, some immune cells are able to modulate angiogenesis [54].

### 3.1. Interferon Family

In humans, there are three subsets of interferon, type I comprising IFNα/β, type II with IFNγ, and also type III with the IFNλs. Type I IFNs are known to inhibit angiogenesis [55], they prevent the production of proangiogenic factors such as bFGF, VEGF, and IL-8 by tumor cells [56]. IFNα/β also inhibits the proliferation of endothelial cells and the secretion of endothelial cell chemotaxis molecules [57]. More specifically, IFNα inhibits the production of bFGF and IL-8 by tumor cells in human bladder cancer cells [58]. Mice deficient for IFNβ show a faster tumor growth in B16F10 and MCA205 cancer models while wild-type mice show better-developed blood vessels. Mice deficient for IFNβ present an increase in neutrophil infiltration and these cells express higher gene-level expressions of VEGF and MMP9 and CXCR4, a neutrophil tumor-homing factor [59]. Type III IFN also has the ability to inhibit angiogenesis [60]. IFNγ is also known to inhibit angiogenesis by inducing angiostasis, the normal regulation system for the creation of new blood vessels [61] (Figure 3).

### 3.2. The Interleukin-1 Family

This family is composed of 11 molecules. Among these, there are agonist ligands such as IL-1α, IL-1β, IL-18, IL-33, IL-36α, IL-36β, IL-36γ and antagonist ligands such as IL-1Ra, IL-36Ra, IL-37 and IL-38. This family is able to mediate angiogenesis indirectly or directly by inducing proangiogenic factors such as VEGF. About 30 years ago, it was shown that IL-1 inhibits endothelial cell growth in vitro and in vivo and is able to inhibit the formation of vessels induced by FGF [62]. In vitro studies have shown that colon, gastric and pancreatic cancer cells can secret IL-1α which in turn enhances angiogenesis [63,64,65]. Studies demonstrated that IL-1α is able to drive angiogenesis in gastric and prostate cancer [66,67]. Melanoma cells are able to secrete IL-1α and IL-1β, which in turn upregulate IL-6, IL-8, the intracellular adhesion molecule-1 (ICAM-1), and the vascular cell adhesion molecule-1 (VCAM-1) expression in endothelial cells [68]. Accordingly, IL-1Ra, which binds to soluble IL-1, reduces angiogenesis [69]. IL-1β promotes tumor growth in a Lewis lung carcinoma model by upregulating VEGF and CXCL2. IL-1β-deficient mice show no local tumor or lung metastases in a B16 melanoma model injected intravenously or intrafootpad [70] (Figure 3).

In the literature, IL-18 is defined as a pro and an anti-angiogenic molecule depending on tissues and context. In the beginning, in vivo studies demonstrated that it negatively regulates neovascularization. In mice subcutaneously injected with T241, IL-18 administration displays an antitumor effect and reduces the microvessel density [71]. Two years later, in vivo and in vitro studies showed that IL-18 can induce endothelial tube formation in rheumatoid arthritis [72]. In a Lewis lung cancer mice model, IL-18 suppresses tumor growth by down-regulating VEGF-A and VEGF-C expression in tumor tissues. It has also been demonstrated that VEGF increases IL-18 production leading to an increase in gastric cancer cell migration [73]. A recent in vivo study showed that macrophage-derived IL-18 inhibits tumor blood vessel formation [74] (Figure 3).

IL-33 is a cytokine with strong angiogenic abilities [75,76,77]. Its receptor ST2 is highly expressed in colorectal cancer cells, stromal cells, and microvessels of colorectal cancers [78]. IL-33 exhibits a proangiogenic effect on Human Umbilical Vein Endothelial Cells (HUVECs) via the Akt pathway [75]. Moreover, IL-33 stimulates myofibroblasts to produce the metalloproteases MMP2 and MMP9, involved in the establishment of new vessels [79,80,81] (Figure 3).

### 3.3. The γc Family

This family is based on their shared expression of the cytokine receptor γ_c._ It is a composed of IL-2, IL-4, IL-7, IL-9, IL-15, and IL-21 [82]. It has been shown that IL-4 is able to block corneal neovascularization through basic fibroblast growth factor and that it inhibits the migration of human microvascular cells [83]. In a spontaneous breast cancer model of mice (PyMT) deficient in IL-4, IL-4 was shown to support vessel remodeling [84]. In NSCLC, the expression of IL-9 is associated with poor prognosis and promotes angiogenesis via STAT3 [85]. IL-15 reduces the mobility of prostate cancer cells and decreases the number of blood vessels in tumor tissue in vivo in mice [86]. Finally, in mice that spontaneously develop intestinal tumors, deficiency in IL-21 reduces angiogenesis [87] (Figure 3).

### 3.4. The Interleukin-6 Family

This family is composed of different members: IL-6, IL-11, ciliary neurotrophic factor (CNTF), leukemia inhibitory factor (LIF), oncostatin M (OSM), cardiotrophin 1 (CTF1), cardiotrophin-like cytokine factor 1 (CLCF1). Recently IL-27, IL-35, and IL-39 have been added to the interleukin-6 family [88].

IL-6 can induce VEGF mRNA expression in A431 cells, a human cell line of epidermoid carcinoma, and also in a rat glioma cell line C6 [89]. Using nude mice, Wei et al. showed that IL-6 promotes tumor growth of a human cervical cancer C33A through VEGF-dependent angiogenesis [90]. In hepatocellular carcinoma, renal cell carcinoma, colorectal cancer, and glioblastoma increased levels of circulating IL-6 are associated with poor response to sunitinib and bevacizumab, a tyrosine kinase inhibitor targeting the VEGF/VEGFR pathway and an anti-VEGF antibody, respectively [89,91,92] (Figure 3).

IL-27 inhibits the production of pro-angiogenic factors by A549 cells, in fact, A549 cells treated with IL-27 show a decrease in VEGF, IL-8/CXCL8, and CXCL5 expression in comparison to non-treated cells. Interestingly, the addition of siRNA against STAT1 increases the levels of these proangiogenic molecules, indicating that IL-27 inhibits the production of angiogenic factors through a STAT1 pathway and VEGF production in human NSCLC [93,94] (Figure 3).

IL-35 is produced in some human cancers such as large B cell lymphoma, nasopharyngeal carcinoma, and melanoma. This interleukin promotes tumor growth by increasing tumor angiogenesis [95]. IL-35 contributes to the progression of prostate cancer through tumor angiogenesis [96] (Figure 3).

### 3.5. The Interleukin-17 Family

This family of pro-angiogenic molecules is composed of six members: IL-17A, IL-17B, IL-17C, IL-17F, IL-17E (also called IL-25), and IL-17F. These molecules bind to IL-17RA, RB, RC, RD, and RE [97]. Patients with colorectal cancer have a poor prognosis if they present a high IL-17 expression which is associated with high microvessel density in colorectal cancer tissues sample [98]. Colorectal carcinoma cell lines express IL-17R and are able to secrete VEGF and IL-6. Interestingly, the stimulation of colorectal carcinoma cells by IL-17 induces the production of angiogenic molecules such as VEGF and IL-6 [99]. Similarly, patients with hepatocellular carcinoma have a poor prognosis when they present an accumulation of Th17 cells in the tumor. The addition of IL-17 in the fibrosarcoma cell line CMS-G4 culture increases the quantity of transcripts of Ang-2 and VEGF [100]. IL-17 also modulates the production of VEGF by an osteosarcoma cell line. In vivo studies show that IL-17A inhibition at the tumor sites suppressed CD31, MMP9, and VEGF expression in tumor tissues [101]. IL-17 is also known to promote resistance to VEGF inhibition therapy [102]. Contrariwise, IL-17F plays a protective role in colon tumorigenesis because IL-17F-deficient mice show an enhanced tumor development, notably with a downregulated angiogenesis in vivo. Accordingly, an in vivo study shows that IL-17F suppresses the tumor growth in mice bearing the hepatocarcinoma cell line SMMC-7721. In the same study, they show that IL-17F inhibits microvessel formation and that it downregulates VEGF, IL-6, and IL-8 expression in hepatocellular carcinoma [103] (Figure 3).

### 3.6. The Interleukin-12 Family

This family includes IL-12 and IL-23. IL-27, IL-35, and IL-39 are also mentioned as members of this family although they are sometimes considered members of the IL-6 family [104]. IL-12R is mostly expressed on activated NK and T cells. In vivo studies showed that IL-12 inhibits angiogenesis. One study showed that NK cell neutralization reduces the ability of IL-12 to inhibit angiogenesis in vivo, suggesting that NK cells mediate the inhibition of angiogenesis by IL-12 [105]. IL-12, by down-regulating angiogenic genes such as CCL2, HIF-1α, VEGF-C, VEGF-D, and IL-6, inhibits the pro-angiogenic activity of human primary lung adenocarcinoma cells [106]. In a murine breast cancer model, IL-12 also inhibits VEGF and MMP9 expression [107] (Figure 3).

### 3.7. The Interleukin-10 Family

This family is composed of IL-10, IL-19, IL-20, IL-22, IL-24, and IL-26 [108]. IL-10 is an anti-angiogenic molecule. Indeed, SCID mice subcutaneously injected with IL-10 transfected B-cells lymphoma DG75 showed a reduced tumor development in comparison with normal cells. The authors showed that these cells inhibit angiogenesis and that in vitro IL-10 inhibits the proliferation of microvascular endothelial cells induced by VEGF and FGF2 [109]. IL-10 also inhibits angiogenesis in mice injected with an ovarian cancer cell line producing VEGF [110]. It was suggested that IL-10 produced by tumor cells inhibits macrophage-derived angiogenic molecules [111]. In NSCLC, IL-20 is potentially anti-angiogenic because it down-regulates COX-2 and VEGF expression [112,113]. IL-22 promotes tumor angiogenesis by stimulating endothelial cell proliferation, survival, and migration. Furthermore, the use of IL-22 neutralizing antibodies inhibits tumor growth, angiogenesis, and microvascular density [114]. The molecule IL-24, in combination with cisplatin, inhibits tumor growth in the xenografted cervical cancer HeLa in nude mice. Furthermore, this combination also inhibits angiogenesis by downregulating VEGF, VEGF-C, and PDGF-B expression [115] (Figure 3).

**Figure 3 cancers-14-02446-f003:**
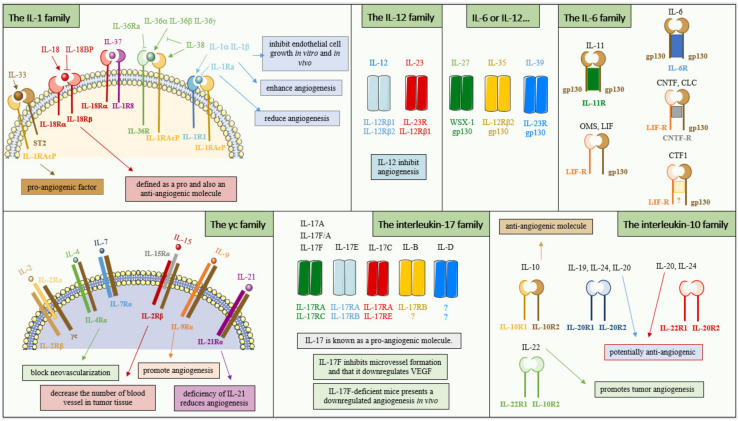
Role of cytokines in angiogenesis.

## 4. Chemokines: Critical Role in Tumor Angiogenesis

Chemokines are divided into four families: C, C-C, C-X-C, and C-X3-C, depending on the arrangement of the two cysteine residues closest to the N-terminal of chemokines. The C chemokines have only one N-terminal cysteine. The C-C chemokines have these cysteines adjacent, the C-X-C chemokines have an amino acid between these cysteines, and the C-X3-C chemokines have three amino acids between these cysteines. These molecules are mainly known to stimulate leukocyte migration but different studies demonstrated that they also play a role in tumor angiogenesis [116,117].

### 4.1. C-C Chemokines

As mentioned above, C-C chemokines have two cysteine residues closest to the N-terminal adjacent. This family is composed of 27 chemokines CCL1 to CCL28, with CCL9 and CCL10 being the same chemokine. These molecules bind to 10 different chemokine receptors, CCR1 to CCR10 [118]. CCL2 is one of the main macrophage chemoattractants and in turn, macrophages recruited by CCL2 secrete proangiogenic molecules such as VEGF. Some patients with glioblastoma multiform (GBM) treated with bevacizumab develop resistance and tumor-associated macrophages notably promote this mechanism. A recent study showed that CCL2 inhibition using mNOX-E38 reduces macrophage migration to CCL2-expressing GBM cells. They also demonstrated that angiogenesis was higher when macrophages and CCL2-expressing cells were cocultured in comparison to CCL2-expressing cells alone. The use of this inhibitor in combination with bevacizumab increases mice survival compared to bevacizumab alone suggesting that CCL2 suppression can increase the efficacy of anti-angiogenic treatments in GBM [119]. In patients with endometrial cancer (EC), the expression of CCL4 and VEGF-A is increased in EC tissues in comparison to healthy individuals, and their expressions are positively correlated. In vitro and in vivo studies demonstrated that CCL4 promotes tumor growth by upregulating the expression of VEGF-A, which affected the STAT3 signaling pathway in EC cells [120]. In human chondrosarcoma, by down-regulating miR-119, CCL5 promotes VEGF-dependent angiogenesis [121]. It also has been shown that CCL11 is able to inhibit angiogenesis by attracting eosinophils in the tumors [122]. In patients with breast cancer, the release of CCL18 by TAMs was positively associated with microvascular density and thus, an increase in angiogenesis. CCL18 also promoted endothelial cell migration and angiogenesis synergistically with VEGF in vitro and in vivo [123].

In CRC tissues, CCL19 is low-expressed in comparison to healthy tissues and CCL19 levels are negatively correlated with angiogenesis. In vitro and in vivo studies show that CCL19 suppresses tumor angiogenesis and that it inhibits angiogenesis in CRC by promoting miR-206 [124]. In vitro and in vivo studies show that the CCL20/CCR6 axis supports angiogenesis. In vitro, CCL20 promotes endothelial cell migration, tube formation, and angiogenesis [125]. In HCC tissues, CCL24 is upregulated in comparison to healthy tissue and is correlated with poor prognosis. CCL24 is able to enhance HUVEC tube formation and also contributes to HCC malignancy through the RhoB-VEGF-A-VEGFR-2 signaling pathway [126]. In an HCC cancer model, hypoxia induces the recruitment of MDSC through CCL26 [127]. In tumors, CCL28 is induced by hypoxia and is able to promote angiogenesis in lung adenocarcinoma by targeting CCR3 in microvascular endothelial cells. In vitro studies demonstrated that this chemokine is able to promote tube formation, proliferation, and migration of endothelial cells [126].

### 4.2. C-X-C Chemokines

This family of chemokines presents angiogenic or anti-angiogenic properties based on the ELR (Glu-Leu-Arg) motif. The presence of this motif promotes angiogenesis and its absence inhibits angiogenesis. CXCL1, CXCL6 and CXCL8, CXCLR5 are ELR-positive and promote angiogenesis whereas CXCL4, CXCL10 and CXCL14 are ELR-negative and inhibit angiogenesis. However, CXCL12 is ELR-negative but promotes angiogenesis. CXCL14 transgenic mice injected with Lewis Lung Carcinoma (LLC) or B16 melanoma cells showed a reduced tumor growth and interestingly, CXCL14 transgenic mice injected with LLC showed a decrease in the number and diameters of visible blood vessels in tumors in comparison to WT mice. Furthermore, the percentage of CD31-positive cells in tumors was higher in WT mice [128]. In human CRC tissues, CXCL5 overexpression is positively correlated with the expression of CD31. This chemokine induces the expression of VEGF-A in HUVEC and is also able to promote HUVEC tube formation, migration, and proliferation through CXCR2 [129]. CXCL1 is also able to promote angiogenesis in colorectal cancer. Interestingly, the receptor of CXCL1, CXCR2 is elevated in CRC tissue and CXCL1 stimulates tumor growth and increases microvessel density [130]. CXCL10 is able to limit the formation of blood vessels by inhibiting endothelial cell migration. CXCL10 inhibits angiogenesis by binding to CXCR3 expressed on newly forming vessels [131]. CXCL8 also known as IL-8 was shown to be an inducer of angiogenesis [132,133,134]. Moreover, a study described CXCL8 as a link between tumor metabolism and angiogenesis. Indeed, in a high-lactate-containing tumor microenvironment, tumor cells can release IL-8 that induces angiogenesis by interacting with endothelial cells [132].

### 4.3. C-X3-C Chemokines

For now, only one chemokine of this family has been described. This molecule is CX3CL1, also known as Fractalkine (FKN), and binds to CX3CR1. This chemokine regulates angiogenesis. Indeed, in vivo and ex vivo studies showed that FKN simulates angiogenesis and in vitro studies showed that this molecule increases proliferation, migration, and tube formation of human umbilical vein endothelial cells. This study showed that CX3CL1 stimulates angiogenesis through the activation of Raf-1/MEK/ERK and PI3K/Akt/eNos signaling pathways [135].

## 5. Non-Classical Pro-Angiogenic Factors

### 5.1. Thymidine Phosphorylase

Thymidine phosphorylase (TP) is an enzyme of the pyrimidine pathway discovered in 1984. This molecule catalyzes the conversion of thymidine to thymine and 2-deoxy-α-D-ribose-1-phosphate. This enzyme is also named the platelet-derived endothelial cell growth factor (PD-ECGF). This molecule is overexpressed in cellular stress conditions such as hypoxia and is expressed by tumoral cells, fibroblast, tumor-associated macrophages, and lymphocytes. TP overexpression is associated with poor clinical outcomes in patients. TP is overexpressed in different cancers such as oral squamous carcinoma, esophageal, gastric, breast, lung, colorectal, bladder, and cervical cancer. This molecule plays an important role in tumor growth by promoting two mechanisms: angiogenesis and apoptosis inhibition. Indeed, TP is an endothelial chemoattractant that stimulates endothelial cell migration as well as angiogenesis factor releases in the tumor microenvironment [136,137,138]. Therapy targeting TP is a promising strategy. First, this enzyme promotes angiogenesis and inhibits apoptosis. Second, it inactivates deoxynucleoside-based therapy and its inhibition may improve the bioavailability of these therapies [137,138,139]. There are different ways to inhibit TP. The first inhibitor developed are pyrimidine and purine analogs such as 6-aminothymine, 6-amino-5-bromouracile, TPI, TAS-102 (TPI and TFT combination), and KIN59. There are also non-nucleobase-based therapies such as: oxadiazole and imidazolidine derivatives, Pyrazalone, and pyrazolo [1,5-a] [1,3,5] triazine analogs, Quinazoline and quinoxaline derivatives, Chromone and isocoumarin derivatives, and finally plant glycosides [139].

### 5.2. Tryptases and Chymases

Tryptase and chymase are pro-angiogenic molecules released from mast cell granules. Tryptase is a tetrameric neutral serine protease while chymase is a monomeric endopeptidase. These two molecules promote directly or indirectly angiogenesis. Tryptase contributes to tube formation and endothelial cell growth by upregulating Ang-1 expression. This molecule induces endothelial cell proliferation, interleukin releases, and in vitro angiogenesis and activates matrix metalloproteinases such as MMP-9 and can convert angiotensin I into angiotensin II. It was also shown that tryptase enhances breast cancer angiogenesis through PAR-2 mediated endothelial progenitor cell activation [140,141,142].

Three classes of tryptase inhibitors have been reported. The first class corresponds to molecules that can form a covalent bond with the catalytic serine in the active pocket of the tryptase. The second class corresponds to molecules containing a basic P1 group that are able to bind to the active pocket of tryptase. The last class of tryptase inhibitors contains molecules with a non-basic P1 group. Some tryptase inhibitors are under clinical trials [143].

## 6. Therapies Targeting Angiogenesis in Cancer

### 6.1. Therapies Targeting the VEGF Family

Therapies targeting the VEGF signaling pathway are the most studied and used in cancer. There are three recombinant proteins approved for cancer treatment: Bevacizumab, Aflibercept, and Ramucirumab. Bevacizumab and Ramucirumab are two humanized monoclonal antibodies targeting, respectively, all VEGF-A isoforms and VEGFR-2. Aflibercept is a protein composed of two recognition domains, VEGFR-1 and VEGFR-2, fused to the Fc portion of a human IgG1. Aflibercept is able to bind to VEGF-A, VEGF-B, and PlGF. There are also tyrosine kinase inhibitors (TKI) approved for cancer: Sorafenib, Sunitinib, Regorafenib, etc. There are many reviews on the anti-VEGF-based therapies, so we decided to not go into much detail. Targeting the VEGF signaling pathway is a promising strategy but, due to many resistances, it appears to be ineffective when used as a single therapy. Indeed, there is a redundancy in the angiogenic signaling pathways, when the VEGF signaling pathway is blocked, other pathways take over to maintain angiogenesis. Therefore, to overcome this resistance, it is of interest to target several angiogenic factors simultaneously.

### 6.2. Theraiesy Targeting Angiopoietin

Therapies targeting the Ang-Tie signaling pathway have recently emerged to treat cancer patients [27]. They are monoclonal antibodies directed against Ang-2 such as MEDI3617, Nesvacumab (REGN910), and LY3127804. MEDI3617 significantly inhibits tumor growth in different xenograft tumor models such as colorectal cancer (LoVo and Colo205), renal cell carcinoma (786-0), ovarian carcinoma (HeyA8), and hepatocellular carcinoma (PLCPRF/5) [144]. Phase I has been achieved to determine its safety in advanced solid tumors (NCT01248949) and another is still under investigation in patients with unresectable Stage III or Stage IV Melanoma (NCT02141542). REGN910 reduces tumor growth and tumor vascularity of different xenograft tumor models such as colorectal cancer (Colo205), prostate cancer (PC3), and epidermoid carcinoma (A431). It also has been shown that REGN910 potentiates the effects of Aflibercept [145]. ABTAA protein is another strategy that not only neutralizes Ang-2 but also activates TIE-2 to enhance vascular normalization and by this, increase drug delivery. This molecule reduces tumor growth in a subcutaneous LLC tumor model [146]. There are also recombinant proteins targeting not only Ang-2 but also the interaction between Ang1/Ang2 with Tie-2. Trebananib is one of these molecules. This molecule is currently under investigation but in combination with paclitaxel, it shows an improved progression-free survival (NCT01204749) for recurrent ovarian cancer. A recent study showed that Bevacizumab plus Trebananib was tolerable and efficient in first-line treatment for patients with metastatic colorectal cancer [147]. There is also the antibody-targeting VEGF-A and Ang-2, called Vanucizumab. However, a recent study demonstrated that the combination of Vanucizumab/mFOLFLOX-6 did not improve the PFS in comparison to Bevacizumab/mFOLFLOX-6 bitherapy in patients with metastatic colorectal carcinoma [148].

### 6.3. Therapies Targeting HGF

Targeting the HGF/MET pathway is a promising strategy because it is involved in different cancer types. There are different ways to target this signaling pathway: HGF inhibitors, MET antagonists, MET kinase inhibitors, and HGF activation inhibitors. MET is expressed in several cell types, including epithelial, endothelial, neuronal, and hematopoietic cells and hepatocytes. The activation of the HGF/MET axis is associated with a series of biological responses, such as proliferation, angiogenesis, migration, invasion, metastasis, and survival. HGF/MET signaling is aberrantly activated in different solid tumors and associated with poor prognosis. HGF/MET aberrant activation plays important roles in the development and progression of several human cancers including lung, renal, gastrointestinal, thyroid, and breast carcinomas, as well as sarcomas and malignancies of the nervous system such as GBM among others. Rilotumumab is a fully-humanized monoclonal antibody targeting HGF. A pre-clinical study showed promising results of a combination of Rilotumumab with docetaxel or temozolomide where Rilotumumab decreases tumor growth in nude mice bearing U-87 MG tumor [149]. In clinical studies, this molecule showed a tolerable profile in patients with mRCC but no effect was identified (NCT00422019). Furthermore, in patients with advanced gastroesophageal adenocarcinoma, there is no benefit to combining Rilotumumab with mFOLFOX6 first-line chemotherapy [150]. In patients with recurrent malignant glioma, Rilotumumab with Bevacizumab did not improve the response in comparison to Bevacizumab alone [151]. For now, the FDA does not accept this molecule.

Onartuzumab is a fully-humanized monoclonal antibody targeting the extracellular domain of MET. In patients with metastatic triple-negative breast cancer, this molecule did not improve the clinical benefit of paclitaxel in bitherapy or not with bevacizumab (NCT01186991). Furthermore, it did not improve the efficiency of mFOLFOX6 in gastric cancer [152]. In patients with metastatic colorectal cancer, the combination of Onartuzumab with mFOLFOX-6 and bevacizumab did not improve the clinical benefit as well [153]. There are two classes of MET tyrosine kinase inhibitors, class I and class II depending on the MET conformation binding [154]. Crizotinib is a type I TKI approved by the European Medical Agency (EMA) and by the Food and Drug Administration (FDA), for patients with NSCLC in particular conditions [155]. This molecule is also approved for patients with anaplastic large cell lymphoma in certain conditions. Cabozantinib is a type II TKI approved by the EMA for patients with medullary thyroid cancer in certain conditions. The FDA has also approved this molecule for patients with locally advanced or metastatic differentiated thyroid cancer.

### 6.4. Therapies Targeting FGF

There are drugs targeting the FGF/FGFR signaling pathway under investigation: monoclonal antibodies targeting FGFR and FGF, and tyrosine kinase inhibitors. Different monoclonal antibodies targeting FGF have shown interesting results. In vivo and in vitro studies showed that antibodies directed against FGF2 and FGF8b have anti-tumor and anti-angiogenic effects [156]. GAL-F2 is a monoclonal antibody targeting FGF2 and was shown to reduce tumor growth in different xenograft mice models of human HCC cell lines: SMMC-7721, HEP-G2, and SK-HEP-1 [157]. Different monoclonal antibodies target FGFR such as FPA144, PRO-001, RG7444, and SSR128129E. There are also antibody-drug conjugates targeting this pathway. The molecule BAY 1187982 is a monoclonal antibody directed against FGFR-IIb and FGFR-IIIc conjugated to a microtubule-disrupting auristatin. This molecule reduces tumor growth in different models such as breast, gastric, and ovarian cancer [158]. There are different tyrosine inhibitors targeting FGFR and also other receptors such as VEGFR, FGFR, PDGFR: AZD4547, BAY1163877, BGJ398, AXL1717, Cediranib, Dovotinib, etc.

### 6.5. Therapies Targeting PDGF

There are different therapies targeting the PDGF/PDGFR signaling pathway: inhibitors of PDGF, inhibitors of the interaction between PDGF and PDGFR, and TKI. There are also human monoclonal antibodies targeting PDGFRα. This molecule reduced tumor growth in a xenograft lung carcinoma tumor model Calu-6 and A549 [159]. It also reduces tumor growth in a xenograft glioblastoma (U118) and leiomyosarcoma (SKLMS-1) [160]. In clinical phases Ib and II, the combination of doxorubicin with an antibody targeting PDGFRα increases the overall survival compared to doxorubicin used alone [161]. This molecule has been approved by the FDA in 2016 for the treatment of soft tissue sarcoma and is under conditional approval by the EMA [162]. In the treatment of glioma, prostate cancer, and ovarian cancer, this molecule is not effective [162]. Finally, there are different TKI targeting the PDGFR signaling pathway clinically approved such as Imatinib, Nilotinib, Dasatinib, Ponatinib, Sunitinib, Axitinib, Sorafenib, etc. [162].

## 7. Conclusions

This review aimed at highlighting the close relationship between angiogenesis and the tumor microenvironment, more specifically the cytokines and chemokines that can be found in tumors. By producing such molecules, cells from the immune system as well as stromal cells, tightly regulate angiogenesis within the tumor.

Inflammation, a key feature of tumorigenesis and a hallmark of cancer, is a strong pro-angiogenic signal. With cytokines such as the IL-1β, IL-6, and TNF all having pro-angiogenic properties, it is clear that inflammation and angiogenesis are related to cancer. Interestingly, cytokines produced by classical pro-tumor immune cells such as Tregs, which produce IL-10 and IL-35; Th17 cells, which produce IL-17 and IL-22; or Th2 cells with IL-4, are all pro-angiogenic factors. On the other side, known antitumor immune cells are linked to anti-angiogenic molecules such as IFNγ and IL-12 with NK cells, Th1, and cytotoxic CD8 T lymphocytes.

Chemokines serve to attract cells in a gradient-dependent manner and their impact on angiogenesis depends on what cells they attract but also on their direct effect on angiogenesis. Indeed, CCL2 will recruit macrophages to the tumor, and induce their production of VEGF as will CCL4. A direct talk has also been found between CXCL1 and tumor cells where CXCL1 induces the production of VEGF by tumor cells. However, some chemokines can also limit angiogenesis by acting directly on newly formed vessels such as CXCL10 or by promoting the expression of the antiangiogenic MiR206 such as CCL19. Chemokines exerting anti-angiogenic effects are usually associated with the recruitment of antitumor immune cells.

There are currently three recombinant proteins targeting the VEGF/VEGFR pathway approved for the treatment of cancer. However, numerous patients develop a resistance to these treatments due to the many redundant pathways leading to angiogenesis. Considerable effort has been made to develop new therapies targeting these redundant pathways with many still in development or under study in clinical trials. It also clearly appears that targeting angiogenesis alone is not sufficient to trigger a potent immune response. Association between anti-angiogenic treatment and chemotherapies or immunotherapies is starting to give promising results and it is likely that more associations of this sort will appear in the future.

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
