# Peer review of "Role of Cytokines and Chemokines in Angiogenesis in a Tumor Context"

_cancers, 2022, doi:10.3390/cancers14102446_

Round 1
Reviewer 1 Report
This is an interesting review focusing on inflammatory cytokines and chemokines in cancer with special regard to angiogenic factors and antiangiogenesis approach. The limit of this review is rapresented by a lot of published review already focusing on this topic. To make the review more appeal and original I suggest to expand it including the discussion of PDEC-GF (Platelet-Derived Endothelial Cell Growt Factor)/ thymidine phosphorylase (TP), tryptase and chymase degranulated from mast cells as a novel non-classical pro-angiogenic factors and the targeting of them. Please I also suggest to expande the reference section with the corresponding references.
Author Response
Reviewer 1
This is an interesting review focusing on inflammatory cytokines and chemokines in cancer with special regard to angiogenic factors and anti-angiogenesis approach. The limit of this review is represented by a lot of published review already focusing on this topic.
Q1
To make the review more appeal and original I suggest to expand it including the discussion of PDEC-GF (Platelet-Derived Endothelial Cell Growt Factor)/ thymidine phosphorylase (TP), tryptase and chymase degranulated from mast cells as a novel non-classical pro-angiogenic factors and the targeting of them. Please I also suggest to expande the reference section with the corresponding references.
Response 1:
We thank the reviewer for this comment and added these information into the manuscript (page 11-12).
- Non-classical pro-angiogenic factors
5.1. Thymidine phosphorylase:
Thymidine phosphorylase (TP) is an enzyme of the pyrimidine pathway discovered in 1984. This molecule catalyzes the conversion of thymidine to thymine and 2-deoxy-α-D-ribose-1-phosphate. This enzyme is also named to platelet-derived endothelial cell growth factor (PD-ECGF). This molecule is overexpressed in cellular stress conditions such as hypoxia and is expressed by tumoral cells, fibroblast, tumor-associated macrophages and lymphocytes. TP overexpression is associated with poor clinical outcome in patient. TP is overexpressed in different cancers such as oral squamous carcinoma, esophageal, gastric, breast, lung, colorectal, bladder and cervical cancer. This molecule plays an important role in tumor growth by promoting two mechanisms: angiogenesis and apoptosis inhibition. Indeed, TP is an endothelial chemoattractant that stimulates endothelial cell migration as well as angiogenesis factor releases in the tumor microenvironnement(Akiyama et al., 2004; Elamin et al., 2016; Liekens et al., 2007). Therapy targeting TP is a promising strategy. First, this enzyme promotes angiogenesis and inhibit apoptosis. Second, it inactivates deoxynucleoside based therapy and its inhibition may improve the bioavailability of these therapies(Bera and Chigurupati, 2016; Elamin et al., 2016; Liekens et al., 2007). There are different ways to inhibits TP. The first inhibitor developed are pyrimidine and purine analogues such as 6-aminothymine, 6-amino-5-bromouracile, TPI, TAS-102 (TPI and TFT combination), and KIN59. There are also non-nucleobase-based therapies such as: oxadiazole and imidazolidine derivatives, Pyrazolone and pyrazolo[1,5-a] [1,3,5] triazine analogues, Quinazoline and quinoxaline derivatives, Chromone and isocoumarin derivatives, and finally plant glycosides(Bera and Chigurupati, 2016).
5.2. Tryptases and Chymases:
Tryptase and chymase are pro-angiogenic molecules releases from mast cells granules. Tryptase is a tetrameric neutral serine protease when chymase is a monomeric endopeptidase. These two molecules promote directly or indirectly angiogenesis. Tryptase contributes to tube formation and endothelial cell growth by upregulating Ang-1 expression. This molecule induces endothelial cells proliferation, interleukins releases and in vitro angiogenesis and activates matrix metalloproteinase such as MMP-9 and can convert angiotensin I into angiotensin II. It was also shown that tryptase enhances breast cancer angiogenesis through PAR-2 mediated endothelial progenitor cell activation (Guo et al., 2016; Qian et al., 2018; Ribatti and Ranieri, 2015).
Three classes of tryptase inhibitors have been reported. The first class corresponds to molecules that can form a covalent bond with the catalytic serine in the active pocket of the tryptase. The second class corresponds to molecule containing a basic P1 group and that are able to bind to the active pocket of tryptase. The last class of tryptase inhibitors contains molecules with a non-basic P1 group. Some tryptase inhibitors are under clinical trials. (Ni et al., 2017).
Reviewer 2 Report
The article is a nice overview of the many growth factors (both classical and immunological) and their role in (tumor) angiogenesis. The extensive figures give a comprehensive overview of it.
-Why is the concept of vasculogenesis not discussed? Not to complete cover this also but it would be good if it is mentioned to exist next to angiogenesis.
-end of paragraph2.1. "..not to develop further" rephrase, this does not cover what you mean. (again on page 10)
-Paragraph 2.1 The paragraph is about VEGF. In the text mainly VEGF-A is mentioned. Sometimes also VEGF, do the authors mean VEGF-A or also the different isoforms? Why is for example VEGF-C not mentioned? Why does PLGF get its own paragraph?
-Chapter 2 is called "the VEGF family" however para's 2.4 and further are about other growth factors. Therefore chapter 2 should be renamed and intro rewritten to introduce all the para's that follow.
-Figure 2, why are the PDGF ligands and receptors given so detailed while a similar (more relevant for this scope) could have been given for VEGF.
-Chapter 3. A general intro to couple the immune system to angiogenesis is missing. It immediately starts with a paragraph on interferon. A general intro here is really needed.
-Chapter 5. A separate paragraph about anti-VEGF therapy would be useful.
-Chapter 5. “Angiopoietin therapy” suggests that the GF is used as a therapy. Anti-angiopoietin therapy would be more clear. Same for the other therapies.
Author Response
Reviewer 2
Q1
Why is the concept of vasculogenesis not discussed? Not to complete cover this also but it would be good if it is mentioned to exist next to angiogenesis.
Response 1:
We tank the reviewer for this comment and added this information in the text (page 1)
“There are two fundamental process to form blood vessels: vasculogenesis and angiogenesis. Vasculogenesis correspond to the de novo blood vessel formation, whereas angiogenesis is the formation of new blood vessel from pre-existing vessels. “
Q2
End of paragraph2.1. "..not to develop further" rephrase, this does not cover what you mean. (again, on page 10)
Response 2:
We changed it by “not to go into details”
Q3
Paragraph 2.1 The paragraph is about VEGF. In the text mainly VEGF-A is mentioned. Sometimes also VEGF, do the authors mean VEGF-A or also the different isoforms? Why is for example VEGF-C not mentioned? Why does PLGF get its own paragraph?
Response 3:
We made the corrections in this paragraph. We decided not to develop VEGFC because its mainly involved in lymphangiogenesis rather than angiogenesis.
Q4
-Chapter 2 is called "the VEGF family" however para's 2.4 and further are about other growth factors. Therefore chapter 2 should be renamed and intro rewritten to introduce all the para's that follow.
Response 4:
We thank the reviewer for this remark and apologize for this mistake that we corrected in the new version.
Moreover, we added an introduction to this paragraph :
“The most important inducer of angiogenesis is the VEGF family. The VEGF family consists of 5 members: VEGF-A, VEGF-B, VEGF-C, VEGF-D and placental growth factor (PlGF). Their biological functions are mediated by 3 receptors: VEGFR-1, VEGFR-2, VEGFR-3 and 2 co-receptors: neuropilin and heparan sulfate proteoglycans. While VEGF-B, PlGF and VEGF-A bind to VEGFR-1, VEGF-A, VEGF-C and VEGF-D bind to VEGFR-2, VEGF-C and VEGF-D bind to VEGFR-3.”
Q5
Figure 2, why are the PDGF ligands and receptors given so detailed while a similar (more relevant for this scope) could have been given for VEGF.
Response 5:
There are already many reviews presenting figures with this information, so we have chosen not to develop here.
Q6
Chapter 3. A general intro to couple the immune system to angiogenesis is missing. It immediately starts with a paragraph on interferon. A general intro here is really needed.
Response 6:
We thank the reviewer for this comment and added the paragraph bellow.
“Angiogenesis is able to modulate the immune system. This mechanism reduces immune cell infiltration by affecting the expression of proteins on endothelial cells. Angiogenesis also induces an immunosuppressive tumor microenvironnement. Indeed, it induces the recruitement of immunosuppressive cells such as Treg and MDSC to the tumor, while it reduces DC maturation and CD3+ proliferation and cytotoxicity. Conversely, some immune cells are able to modulate angiogenesis (Geindreau M et al. 2021). “
Q7
Chapter 5. A separate paragraph about anti-VEGF therapy would be useful.
Chapter 5. “Angiopoietin therapy” suggests that the GF is used as a therapy. Anti-angiopoietin therapy would be more clear. Same for the other therapies.
Response 7:
Again we thank the reviewer for these suggestions and modified the text.
Round 2
Reviewer 1 Report
The review is now improved. The new added references in the main text need to be inserted in the References section
Author Response
We apologize for the issue raised by the reviewer. We have made the necessary corrections.Reviewer 2 Report
The text is written much better now and all points are resolved, well done!
Author Response
Thank you.
Round 3
Reviewer 1 Report
The manuscript is now suitable for publication